# Membrane Curvature: The Inseparable Companion of Autophagy

**DOI:** 10.3390/cells12081132

**Published:** 2023-04-11

**Authors:** Lei Liu, Yu Tang, Zijuan Zhou, Yuan Huang, Rui Zhang, Hao Lyu, Shuai Xiao, Dong Guo, Declan William Ali, Marek Michalak, Xing-Zhen Chen, Cefan Zhou, Jingfeng Tang

**Affiliations:** 1Cooperative Innovation Center of Industrial Fermentation (Ministry of Education & Hubei Province), Hubei Key Laboratory of Industrial Microbiology, Hubei University of Technology, Wuhan 430068, China; 2National “111” Center for Cellular Regulation and Molecular Pharmaceutics, Key Laboratory of Fermentation Engineering (Ministry of Education), Hubei University of Technology, Wuhan 430068, China; 3Department of Biological Sciences, University of Alberta, Edmonton, AB T6G 2R3, Canada; 4Department of Biochemistry, University of Alberta, Edmonton, AB T6G 2R3, Canada; 5Membrane Protein Disease Research Group, Department of Physiology, Faculty of Medicine and Dentistry, University of Alberta, Edmonton, AB T6G 2R3, Canada

**Keywords:** autophagy, membrane curvature, Atg proteins, ER-phagy, nucleophagy, xenophagy

## Abstract

Autophagy is a highly conserved recycling process of eukaryotic cells that degrades protein aggregates or damaged organelles with the participation of autophagy-related proteins. Membrane bending is a key step in autophagosome membrane formation and nucleation. A variety of autophagy-related proteins (ATGs) are needed to sense and generate membrane curvature, which then complete the membrane remodeling process. The Atg1 complex, Atg2-Atg18 complex, Vps34 complex, Atg12-Atg5 conjugation system, Atg8-phosphatidylethanolamine conjugation system, and transmembrane protein Atg9 promote the production of autophagosomal membranes directly or indirectly through their specific structures to alter membrane curvature. There are three common mechanisms to explain the change in membrane curvature. For example, the BAR domain of Bif-1 senses and tethers Atg9 vesicles to change the membrane curvature of the isolation membrane (IM), and the Atg9 vesicles are reported as a source of the IM in the autophagy process. The amphiphilic helix of Bif-1 inserts directly into the phospholipid bilayer, causing membrane asymmetry, and thus changing the membrane curvature of the IM. Atg2 forms a pathway for lipid transport from the endoplasmic reticulum to the IM, and this pathway also contributes to the formation of the IM. In this review, we introduce the phenomena and causes of membrane curvature changes in the process of macroautophagy, and the mechanisms of ATGs in membrane curvature and autophagosome membrane formation.

## 1. Introduction

The biological membrane is one of the basic structures of the cell. It consists of a phospholipid bilayer that acts as a physical barrier to separate distinct intracellular compartments (such as the endoplasmic reticulum (ER), Golgi apparatus, and nucleus), and separates the cell from the external environment [1]. Membranes are highly dynamic and can therefore form a variety of complex shapes. It was found that the membrane shape is closely linked to cellular function in the process of membrane transport [2]. There are a large number of membrane structures in cell organelles that are closely related to their functions [3]. For example, the ER has two shapes: sheet and tubular. ER sheets are the site of insertion of membrane proteins into the membrane, whereas ER tubules regulate lipid synthesis and ion homeostasis [4,5], and perform their functions by forming membrane contact sites (MCSs) with other organelles [6]. By contrast, the Golgi is composed of flattened vesicles of different sizes, and these vesicles work in concert to process and translocate proteins. The function of these complex membrane structures inevitably leads to membrane deformation, and thus the generation of membrane curvature is essential for cellular activity. It was found that membrane curvature plays a key role in many cellular life activities, such as membrane transport, membrane fusion, vesicle transport, and mitosis [7]. In recent years, studies have demonstrated that membrane curvature also plays an important function in the formation of autophagosomal membranes in autophagy [3].

Autophagy is an important mechanism for cellular homeostasis, and protects cellular function from damage. It is also an evolutionarily conserved intracellular degradation process [8]. There are three main types of autophagy: macroautophagy, microautophagy, and molecular chaperone-mediated autophagy (CMA). In macroautophagy, the cytoplasmic cargo is transported to the lysosome/vacuole via a double-membrane vesicle called the autophagosome. The autophagosome binds to the lysosome/vacuole to degrade the cargo by acidic hydrolases. In microautophagy, the lysosomal membrane can invaginate to form vesicles containing cytoplasmic cargo. The vesicles are degraded in lysosomes [9]. In CMA, the chaperones specifically identify the proteins with the KFERQ motif and target them to the lysosome membrane. Then, the proteins are transported to lysosomes by receptor proteins [10].

In autophagy, various ATGs regulate the autophagic process by interacting with the membrane. Among these proteins, six complexes are essential for the formation of the phagophore in starvation-induced autophagy (Figure 1a) [11], namely the Atg1 complex, Atg2-Atg18 complex, Vps34 complex, Atg12-Atg5 conjugation system, Atg8-phosphatidylethanolamine (PE) conjugation system and transmembrane protein Atg9. The Atg1 complex acts as a scaffold for the phagophore assembly site, also called the pre-autophagosomal structure (PAS) and recruits Atg9 vesicles to form the isolated membrane (IM) precursor. The Vps34 complex produces phosphatidylinositol 3-phosphate (PI3P) and recruits the Atg2-Atg18 complex and two ubiquitination systems. In this process, the IM continues extending, and eventually forms a mature autophagosome [12]. In this review, we will summarize the changes in membrane curvature in the process of membrane formation and elongation, and present several examples in selective autophagy.

## 2. Membrane Curvature

Membrane curvature is the result of complex interactions between lipids, membrane proteins, and physical forces exerted on the membrane surface [13]. The phospholipid bilayer is the basic scaffold of the cell membrane, which is made of many lipids. Due to the strong negative spontaneous curvature of the lipid cholesterol, fatty acid, and diacylglycerol on the membrane, it is possible to generate a large membrane curvature. Some membrane proteins are partially or completely inserted into the phospholipid bilayer to directly alter the membrane curvature. Other proteins are attached to the surface of the membrane to change the membrane curvature by applying forces to the membrane surfaces [14].

There are three mechanisms to explain how proteins generate membrane curvature by applying forces to the membrane surface (Figure 1b). The first mechanism is the scaffolding mechanism which postulates that proteins act as scaffolds. These proteins present the intrinsic curvature of proteins to the phospholipid bilayer, causing the local bilayer to bend [15]. For example, the BAR domain has a crescent-like shape, and its concave surface binds to the lipid membrane. The protein has some positively charged residues on its concave surface, which allows it to interact strongly with the negatively charged lipid molecules on the membrane. Then, the curvature of the membrane in this area becomes close to the curvature of the BAR domain [16]. The BAR domain family plays a vital role in many cellular activities, such as endocytosis and cytokinesis. It is can also connect the plasma membrane to the actin cytoskeleton. As a scaffold, the BAR domain is very conserved. There are three helical structures, which normally form curved homo- or heterodimers with a crescent-shape. The BAR domains are classified into the classical BAR domain, Fer/Cip4 homology (FCH) BAR domain (F-BAR), and inverse BAR domain (I-BAR) proteins. The classical BAR domain has a higher intrinsic curvature, such as in the amphiphilic *N*-terminal helix protein (N-BAR), that can promote positive membrane curvature through its intrinsic curvature [17]. The intrinsic curvature of the F-BAR domain proteins ranges from high to low. Therefore, it can support a large range of positive membrane curvatures [18]. The I-BAR domain proteins have no intrinsic curvature and are associated with a negative curvature. This enables the creation of scaffolds on flat membrane surfaces [19,20]. The second mechanism is called the local spontaneous curvature mechanism [13]. This mechanism is based on the local deformation of the membrane that occurs when the amphiphilic portion of a protein is embedded in the lipid matrix. The shallow insertion of amphiphilic protein helices into the upper part of the membrane monolayer interferes with the accumulation of lipid polar head groups and leads to local monolayer deformation. The third mechanism is one in which proteins can change the shape of the membrane according to a bilayer coupling mechanism [13]. If amphiphilic protein penetrates only one lipid monolayer, it can create differences in area between membrane leaflets. The membrane will form a curvature to compensate for this area asymmetry. However, this difference has an almost negligible effect on the curvature of the membrane compared to the membrane area of the entire membrane surface area [13].

## 3. Membrane Curvature in the Initiation Phase of Autophagy

It is known that autophagy is an adaptive process that responds to different forms of stress, including nutrient deprivation, growth factor depletion, infection, and hypoxia. The initiation of autophagy in yeast occurs in the PAS, and the beginning of mammalian autophagy occurs in specific domains of the ER called omegasomes that are located in the tubular ER [21].

The ER is the largest membrane-bound organelle in eukaryotic cells, composed of the nuclear envelope and peripheral ER. It is capable of performing a variety of essential cellular functions, including protein synthesis and processing, lipid synthesis, and calcium (Ca^2+^) storage and release [22]. The nuclear envelope consists of two lipid bilayers that share a tubular lumen with the peripheral ER. The peripheral ER consists of two structures: sheet ER and tubular ER. The sheet ER, which is the site of synthesis, folding, and translation of membrane and secretory proteins, is covered with a large number of ribosomes [23]. In contrast, the surface of the tubular ER is highly curved and smooth, with only a small distribution of ribosomes, and participates in lipid synthesis, Ca^2+^ signaling, and uncoupling from other organelles [24]. The tubule shape of the ER is maintained by two classes of evolutionarily conserved proteins including Rtns and REEPs. In the absence of these proteins, the tubules are converted to sheets. These tubule proteins contain four transmembrane (TM) domains and an amphipathic helix (APH) at the C-terminus. It was found that the dimerization of REEP5 is mediated by TM2 and that the APH promotes dimer formation [25]. The insertion of REEP5 into liposomes enables the conversion of phospholipid bilayers into lipoprotein particles (LPP) with a higher curvature, whereas APH also promotes the formation of LPP. The dimer of REEP5 spans the phospholipid bilayer in a V-shape, leading to membrane asymmetry and generating local membrane curvature, whereas APH inserts directly into the phospholipid bilayer generating local membrane curvature.

On the tubule ER, the initiation of autophagy depends on the synthesis of the IM, which gradually absorbs phospholipids. The Atg1 and Vps34 complexes play important roles in this process. The Atg1 complex is the basis of the PAS, and the Vps34 complex recruits the PI3P, which is a component of the membrane.

### 3.1. Atg1 Complex

The Atg1 complex acts as a scaffold for the PAS, recruiting downstream ATGs and controlling the IM formation. The Atg1 complex is composed of Atg1, Atg13, Atg17 (mammalian homolog FIP200), Atg31, and Atg29. Atg17 plays an important role in tethering vesicles because of its structural peculiarities. Atg17 consists of four helices and has an overall crescent-shaped structure, which is similar to the BAR domain that binds to tubular vesicles [26]. Based on the crescent-shaped structure, Atg17 is predicted to bind to curved membranes. However, the Atg17 is not predicted to appear crescentic based on the predicted amino acid sequence, and there is no sequence homology between Atg17 and any other BAR domain. Atg17 is a monomer forming a crescentic structure with two tips that are not symmetrical, whereas the BAR domain functions as a dimer and usually appears in a symmetrical form [27]. These phenomena suggest that Atg17 and the BAR domain bind to the membrane and affect membrane curvature through different mechanisms.

It has been shown that Atg17 functions as a PAS protein, usually forming a complex with Atg29 and Atg31 in a ratio of 2:2:2 [28]. Atg31 contains a β-sheet and an α-helix structure [29], and binds to the *N*-terminal of Atg29 through its *N*-terminal β-sheet and to Atg17 through its C-terminal helix structure. In addition, there is no direct binding between Atg17 and Atg29, and instead Atg31 acts as a bridge connecting Atg17 and Atg29 [30,31]. The Atg29-Atg31 complex locates at the center of the Atg17 crescent and regulates the binding of the crescent structure and bent vesicles, whereas Atg17 functions as a scaffold [32]. The dimerization of Atg17-Atg31-Atg29 is mediated by the C-terminal helix of Atg17, and the dimer presents a crescent shape. Based on the structure, some scientists speculated that the dimer might act as a lipid vesicle curvature sensor [31]. 

In the yeast model, Atg13 is the substrate of Tor1. Under nutrient-rich conditions, the C-terminus of Atg13 is highly phosphorylated. At the same time, the binding between Atg13 and Atg1 is weakened [33]. Under starvation or rapamycin (Tor1 inhibitor) treatment, Atg13 is rapidly dephosphorylated [34], and the binding of Atg13 and Atg1 is strengthened at the PAS. Through the interaction of Atg13 and Atg17, the Atg1-Atg13-Atg17-Atg31-Atg29 complex is assembled, which subsequently initiates autophagy [35]. It was shown that the Atg1 complex is involved in membrane targeting in vivo [36]. The EAT domain of Atg1 prefers to bind to vesicles and also contains a binding site to Atg13, suggesting the EAT domain is essential for the formation of the Atg1 complex and targeting to the membrane [31]. Based on this structure, a model is proposed for explaining how the Atg1 complex binds Atg9 vesicles in the formation of the IM [37]. The Atg29-Atg31 complex, located at the center of Atg17, has a spatial site block and is partially mobile relative to Atg17. The special structure of Atg29-Atg31 prevents vesicles from remaining in the crescent-shaped domain [31]. Under starvation conditions, Atg1-Atg13 associates with Atg17 at the APS to initiate autophagy, and Atg29-Atg31 gates vesicle access to the Atg17 crescent (Figure 2a) [37]. Because Atg1 and Atg13 sense curvature and bind Atg17, they may restrict the localization of Atg17 and Atg9 at the phagophore [37].

In the mammalian model, ATG13 (the mammalian homolog of Atg13) binds ULK1/2 (the mammalian homolog of Atg1) and mediates the interaction of ULK1/2 and FIP200 [38]. ATG10 maintains the stability of the ULK1/2 complex by forming heterodimers with ATG13, for which there is no homolog in yeast [39]. In the presence of amino acids, the mTOR phosphorylates ATG13 and ULK1/2 to inhibit autophagy. Upon amino acid deprivation, the activity of mTOR is inhibited [40]. The dephosphorylation of ULK1 and ATG13 results in the activation of ULK1 kinase and the induction of autophagy [41]. It was shown that the ULK-ATG13-FIP200 complexes are direct targets of mTOR and important regulators of autophagy in response to mTOR signaling [42].

### 3.2. Vps34 Complex

The initiation phase of autophagy is inseparable from the involvement of the Vps34 complex, which consists of the core components Vps34 (mammalian homolog VPS34), Vps15, and Beclin1. It is divided into complex I and II according to the specific components Atg14 (mammalian homolog ATG14L) and Vps38 (mammalian homolog UVRAG) (Figure 2b) [43,44]. Vps34 phosphorylates phosphatidylinositol (PI) to produce PI3P, which affects the extension of the IM and the recruitment of ATGs to vesicles. Vps34 is activated by binding to Vps15 and further binds Beclin1 to form the Vps34-Vps15-Beclin1 complex, which forms the core component of the Vps34 complex [45]. During starvation-induced autophagy, Vps34 complex I functions at the IM and ER [46,47,48], whereas complex II functions in the endocytic pathway by co-localizing with Rab5 and Rab7 on lysosomes in mammals and the transit between late endosomes and in the trans-Golgi network by localizing on Rab9 [46].

The binding of proteins to the membrane is influenced by three important physicochemical parameters: membrane electrostatic forces, lipid packing, and membrane curvature [49]. The source of the negative charge of the cell membrane is mainly the phosphate group on phospholipids in the membrane and glycosylation modifications of the extracellular domain of cellular proteins. Lipid packing is determined by the shape of the polar lipid head group and the degree of saturation of the acyl chain, and the saturation state of the acyl chain also affects membrane curvature. Thus, a combination of factors affects protein-membrane binding [49]. It was shown that lipid packing can affect the saturated acyl chain and inhibit the activity of the two Vps34 complexes, but increasing membrane curvature backfilled the activity of the complexes, which in turn affected the autophagic process [50]. In addition, complex components Bif-1 and Atg14 also affect membrane curvature, and thus autophagy, directly or indirectly through interactions.

#### 3.2.1. Bif-1

Bif-1 (Bax-interacting factor 1), also known as endophilin B1, is both a member of the endophilin family and a Bax-binding protein [51,52]. Bif-1 is localized in the cytoplasmic matrix and regulates the membrane dynamics of organelles including the Golgi complex, mitochondria [53], and autophagosome [54]. Bif-1 contains an *N*-terminal BAR domain and a C-terminal SH3 domain [55]. The BAR domain consists of three reverse parallel helices which are necessary for binding the lipid bilayer and inducing membrane bending (Figure 2b). These helices form a helical coil and allow the BAR domain to dimerize. The BAR domain curvature sensing depends on the formation of this dimer [56].

It was hypothesized by scientists that the *N*-terminal α amphiphilic helix (H0) on Bif-1 can anchor curvature-producing proteins on the membrane and insert them into the membrane, promoting curvature [57]. This hypothesis has been demonstrated using electron paramagnetic resonance spectroscopy (EPR) in which the α helicity of the N-BAR domain increased from 36% to 48% when bound to the membrane [55]. The helices on hydrophobic and hydrophilic amino acid residues are distributed on both sides of the helix wheel, and hydrophobic amino acid residues are inserted into the membrane protein, generating membrane curvature [55]. The concave side of the BAR domain is positively charged and regulates membrane curvature by binding to negatively charged membranes like a scaffold. In addition, there is an insertion amphipathic helix (H1) of the BAR domain that inserts into the membrane. The H1 affects the dimerization of the BAR domain and promotes the generation of membrane curvature [55]. 

It was shown that treatment with EBSS induced a large number of autophagic vacuoles in wild cells [58]. In knockout (KO) SH3GLB (encoding Bif-1), the number of autophagic vacuoles was significantly reduced, suggesting that Bif-1 is involved in the formation or processing of autophagic vacuoles [58]. Meanwhile, KO SH3GLB inhibited the formation of LC3 spots. Overexpression of Bif-1 in cells with knockdown SH3GLB restored the expression of LC3 spots, but overexpression of SH3GLB ΔBAR mutants did not restore the expression of LC3 spots, indicating that the BAR domain is required for Bif-1-induced IM formation [59]. 

UVRAG contains an *N*-terminal proline rich (PR) sequence, a calcium-dependent phospholipid-binding domain (C2), and a central convoluted domain (CCD). It was shown that UVRAG directly interacted with the Beclin1-VPS34 complex through its CCD domain to activate autophagy [58], whereas Bif-1 also bound to UVRAG through its SH3 domain, suggesting that UVRAG may be a “bridge” between Bif-1 and Beclin1 binding [59]. Based on the above findings, a hypothesis was proposed. During starvation-induced autophagy, Bif-1 regulates the fission of Golgi membranes. Bif-1 together with the UVRAG-Beclin1-VPS34 complex may regulate the formation of Atg9 vesicles at the Golgi complex, and Atg9 is trafficked to the IM for its expansion [54,56,59]. 

#### 3.2.2. Barkor/Atg14

Atg14 primarily localizes in the initiating IM and co-localizes with the early autophagy marker protein Atg16L1 and the omegasome marker protein DFCP1 [60]. Atg14 contains an *N*-terminal zinc finger and a CCD domain (Figure 2b). It binds to the CDD domain of Beclin1 through its own CDD domain, localizing Beclin1 to the IM and thus promoting autophagy [45]. In addition, UVRAG binds to Beclin1 through the CCD domain, suggesting that both of them competitively bind to Beclin1 to affect autophagy.

It was shown that the BATS domain, which consists of the last 80 amino acids at the C- terminal of Atg14, is the smallest region that localizes to the IM. The BATS domain targets Beclin1 to the IM and activates autophagy, and 19 amino acids in this domain form a classical amphiphilic α-helix, in which hydrophilic amino acids arrange on either side of the helices and hydrophobic residues embed in the lipid bilayer. The BATS domain acts as a curvature sensor to bind to the highly curved early IM and preferentially binds to membranes with PI3P and PI(4,5)P [61].

In summary, Atg14 is targeted to the ER membrane via its *N*-terminal cysteine repeat sequence [62]. In autophagic stress, the Beclin1 complex is recruited to the ER, where PI3P is produced to support the curvature of the IM [63]. The high curvature membrane further attracts Atg14, which leads to more PI3P production and promotes the autophagic process due to the high affinity of its C-terminal BATS domain for the high curvature membrane [62].

### 3.3. Atg9

Atg9 (mammalian homolog ATG9A) is the sole multiple transmembrane protein in autophagy with four transmembrane helices and two helices buried in the cytoplasmic leaflet [64]. Atg9 is predicted to cause membrane bending, and localizes at highly curved membranes, including vesicles and the expanding edge of the IM [64]. When autophagy is induced, the vesicles, which contain Atg9, are sorted from the Golgi/ endosome and translocated to the PAS [65]. At the PAS, the Atg9 vesicles are tethered by the Atg17 of the Atg1 complex, and three vesicles tether and fuse to become a part of the IM, becoming one of the initial membrane sources for the IM (Figure 2a) [66]. 

In the extension phase of the initial IM, Atg9 is relocated to the expanding edge of the IM together with the Atg2-Atg18 complex as the IM expands (Figure 2c). Before the IM forms a complete autophagosome, the Atg2-Atg18 complex detaches from the IM while Atg9 remains on the completed autophagosome [67]. Atg2 transfers phospholipids from the ER to cytoplasmic leaflets of the membrane, and these phospholipids are then distributed to the luminal leaflet for membrane expansion [68,69]. Lipid transport between the two leaflets of the membrane is mediated by lipid translocases, including flippases, floppases, and scramblases [70,71]. Flippases and floppases require ATP to transport lipids in a unidirectional manner, whereas scramblases do not require ATP to transport lipids in a bidirectional manner. It was shown that Atg9 is a lipid scramblase, and transports lipids between the two leaflets of the membrane for IM expansion [72].

In addition to the ER, the IM contacts with other organelles, such as the nuclear membrane or lipid droplets (LDs), are also crucial for autophagosome biogenesis [73]. For starvation-induced autophagy in yeast, LDs are necessary [74]. The absence of LDs inhibits the formation of the autophagic structure, and LDs have been proposed to act as a source of lipids for the growth of the IM [75]. LDs regulate autophagy by contributing to ER homeostasis as well as maintaining the phospholipid composition [76].

## 4. Membrane Curvature in the Extension Phase of Autophagy

The extension of the IM cannot be achieved without the involvement of phospholipids. There are three models to explain the origin of phospholipids. The first is a direct connection model [77], in which the ER binds to the IM, and phospholipids are transferred directly from the ER to the IM. The second is a vesicle-mediated model [78], in which vesicles from the ER or other membrane organelles fuse with the IM to provide phospholipids to the IM. The third is an Atg2-mediated model [79], in which phospholipids from the ER are transferred to the IM via Atg2.

### 4.1. Atg2-Atg18 Complex

ATG2 is a member of the lipid transport proteins that are required for early IM formation [68], and it mediates the transfer of lipids from the ER, the site where synthesis occurs, to the IM (Figure 2c). There are two membrane-binding domains in the *N*- and C-terminal regions of Atg2. The C-terminal amphipathic helix facilitates Atg18 binding to PI3P and thus targets the Atg2-Atg18 complex to the PAS. Meanwhile, the *N*-terminal associates with the ER to form the IM [80]. In fact, Atg2 is required to establish membrane contact sites between the IM and the ER [67,69,80]. Protein-mediated lipid transfer occurs exclusively between the cytosolic leaflets, but not luminal leaflets of apposed organellar membranes at membrane contact sites [81]. Atg2, together with Atg9 and Atg18, is located on the edges of the IM, which are highly-curved regions, but also preferentially associates with membranes carrying lipid-packing defects [67,82]. Atg2 forms a hydrophobic "bridge" between the ER and the IM, along which lipids flow from the ER to the cytoplasmic leaflets of the IM, while blocking the return of lipids from the IM to the ER [83]. Recently, it was shown that Atg9 is a scramblase protein, allowing the asymmetry in the lipid gradient to transport lipids between the two leaflets of the membrane for IM expansion [72,84,85]. Left unchecked, either lipid depletion in the ER or lipid enhancement in the IM will lead to asymmetry in the bilayer [85].

### 4.2. Atg8-PE and Atg12-Atg5 Conjugation System

The Atg8 ubiquitin-like protein family encoded in the human genome consists of seven members, including LC3A/B/B2/C and GABARAP/L1/L2, which play an important role in the autophagic process. Lipidation of the Atg8 protein is a key event in the process of cellular autophagy. Only the completely lipidated Atg8 protein can bind to the IM and participate in the autophagic process [86]. The lipidation reaction is catalyzed by the E1-E2-E3 enzymatic cascade reaction (Figure 2d) [87]. After Atg8 is synthesized, its carboxyl terminus is sheared by Atg4 to expose glycine residues to the cytoplasm [88]. Atg8 is activated by binding to the E1 ubiquitin-activating enzyme Atg7 using ATP, and is then transferred to E2 ubiquitin-binding enzyme Atg3, forming a thioester-bonded Atg3~Atg8 intermediate [36]. As an E3 ubiquitin ligase, the Atg12-Atg5-Atg16 (mammalian homolog ATG12-ATG5-ATG16L1) complex interacts with intermediates to facilitate the transfer of Atg8 from the cysteine of Atg3 (mammalian homolog ATG3) to the head of phosphatidyl ethanolamine (PE) lipids, and the Atg8 is attached to the IM in a membrane-bound form [87,88,89,90,91]. In this process, the ATG3 and ATG12–ATG5-ATG16L1 complexes are both involved in the conjugation of ATG8 proteins to membranes [92,93].

The Atg3-Atg8 complex is generated regardless of the presence of liposomes, and therefore the transfer from Atg3 to PE is a membrane curvature-sensitive step, and Atg3 may be a component of the curvature sensor. It was shown that when the content of DOPE in the liposome was high, the binding efficiency of Atg3 to the membrane was not affected by the curvature [93]. However, when the content of DOPE in liposomes was low, the binding efficiency of Atg3 to membranes was very dependent on the curvature. The higher the curvature, the higher the binding efficiency, indicating that Atg3 bound to the membrane in a curvature-dependent manner. Moreover, when the curvature was sufficiently high, Atg3 bound to the membrane even if the liposome excluded DOPE, indicating that Atg3 has a curvature-sensitive membrane-binding motif [93].

There are two main models for sensing curvature: one in which the BAR domain relies on its own membrane curvature, and the other in which it is inserted into the phospholipid bilayer through an amphiphilic helix [94]. Atg3 contains an *N*-terminal amphipathic helix, an intrinsically disordered flexible region (FR) bound to Atg7, a handle region (HR) consisting of the helix, and a disordered region bound to Atg8 and a flexible C-terminus. Three segments compose the catalytic core domain (Cat), interspersed in Atg3 to function as a ubiquitin-binding enzyme, which includes a long central helix and two catalytic segments [95,96]. The E123IR domain of the FR region is bound to the *N*-terminal NTD domain of Atg7 [97]. The E123IR domain is a metastable switch. In the absence of the enzymatic cascade, this domain interacts intramolecularly with the Cat domain of Atg3. When both E1 and E2 enzymes are present, the structure of Atg3 is rearranged, and subsequently Atg3 is activated to induce Atg8 lipidation [2].

The 27 residues at the *N*-terminal of ATG3, named NT, contain a variable region (NT^Var^) and a conserved region (NT^Cons^) [98]. The NT^Var^ that makes up most of the ATG3 amphipathic helix is responsible for the membrane curvature-dependent binding. Although the NT^Var^ of Atg3 shares poor sequence similarity with ATG3, they all are sensitive to membrane curvature but are not functionally interchangeable. The NT^Var^ is necessary for targeting ATG3 to the membrane [99] but is not sufficient for LC3-PE conjugation [98]. The NT^Cons^ of ATG3 is key to structural rearrangements that are caused by the binging of *N*-terminal amphipathic helices to the membrane. The NT^Cons^ of ATG3 works with the amphipathic helix of NT^Var^ to promote LC3-PE conjugation [98].

It was shown that the ATG12-ATG5-ATG16L1 complex as a dimer stabilized the phagophore rim via a preference for high curvature membrane binding [92]. WIPI2 is a member of the PROPPIN family, which binds highly curved membranes through a hydrophobic loop and an FRRG motif. The FRRG motif specifically recognizes PI3P [100]. There are two amphipathic helices near the *N*-terminal of ATG16L1, and the helix α2 is responsible for curvature sensitivity. The specific motif in the C-terminal of ATG16L1 also binds directly to membranes [92]. WIPI2 is required to recruit ATG12-ATG5-ATG16L1 to flat membranes [101], and the ATG12-ATG5-ATG16L1 complex induces membrane curvature at a high surface density [102] (Figure 2d). It was demonstrated that ATG16L1 and WIPI2 curvature sensing was independent of ATG3 [102]. However, the properties of yeast Atg16 have been reported to differ from mammalian ATG16L1, so the model may be not suitable for yeast Atg16 [102].

## 5. Membrane Curvature in the Maturation Phase of Autophagy

The mechanism of fusion between the autophagosome and lysosome is unknown during autophagy maturation. It was reported that the fusion process is regulated by ATG14 [103,104]. The CCD domain of ATG14 interacts with the α helix of the STX17 SNARE motif, and the interaction requires ATG14 homo-oligomerization through cysteine repeats [103]. The binding of ATG14 and the STX17-SNAP29 complex promotes STX17-SNAP29-VAMP8-mediated autophagosome fusion with lysosomes [104]. However, the homo-oligomerization of ATG14 is required for fusion, whereas not for the formation of the IM [104]. 

Autolysosomes contain components of the IM and lysosomes. The autophagosome inner membrane and enclosed substrates are degraded, and the autophagosome outer membrane components are reported to be recycled by a recycler composed of SNX4, SNX5, and SNX17 [105]. The sorting nexins (SNXs) are a family of peripheral membrane proteins that control trafficking within the endocytic network [106]. In autophagosomal components recycling, SNX4 and SNX5 form a heterodimer. The heterodimer recognizes autophagosomal membrane proteins and generates membrane curvature on autolysosomes via their BAR domain to mediate the cargo sorting process [105]. SNX17 interacts with both the dynein-dynactin complex and the SNX4-SNX5 dimer to facilitate the cycle of autophagosomal membrane components [105].

## 6. Membrane Curvature in Selective Autophagy

### 6.1. ER-Phagy

ER-phagy is a type of selective autophagy. When the ER is subjected to external stimuli leading to functional changes, the damaged ER is removed by selective autophagic degradation to restore the remaining ER. Six ER resident proteins have been identified to act as autophagy receptors: FAM134B [107], SEC62 [108], RTN3L [109], CCPG1 [110], ATL3 [111], and TEX264 [112]; two ER-phagy receptors were also identified in yeast, namely Atg39 and Atg40 [113,114]. These autophagy receptors bind to the Atg8 family via the ATG8-interacting motif (AIM) W/F/Y-X-X-L/I/V. This binding allows the IM to effectively target and degrade the target protein.

FAM134B, the first ER-phagy receptor identified, contains a reticular homology domain (RHD) and an AIM domain. The RHD domain contains two large transmembrane domains (TM12 and TM34) anchored to the ER membrane, and two amphipathic helices positioned on the membrane on either side of the TM34 domain (Figure 3a). The insertion of the RHD domain disrupts the symmetry of the bilayer. The amphipathic helices deform the bilayer by stretching the cytoplasmic leaflets of the membrane to induce membrane curvature [115] and complete the remodeling process of part of the ER membrane. FAM134B binds to ATG8 through its AIM domain to recruit the IM and promote ER-phagy (Figure 3b). The FAM134 reticulon family proteins (FAM134A/B/C) have a similar structure and function, and all act as receptors [116]. However, there are some differences. For example, the overexpression of FAM134B was sufficient to promote ER-phagy [117]. FAM134A/C exhibits a limited ability to fragment ER under basal conditions, which was augmented by environmental stresses [118,119]. 

Atg40 [120] contains a reticulon-like domain and an AIM domain (Figure 3a). The reticulon-like domain induces the production of highly curved ER fragments, and the AIM domain interacts with Atg8 on the IM to incorporate these fragments into the IM to promote ER-phagy (Figure 3b).

### 6.2. Nucleophagy

Nucleophagy, which clears damaged nuclear components to maintain nuclear integrity, is the degradation of nuclear material including nuclear membrane, nuclear lamina, nucleoplasm, nucleolus, and DNA [121]. Atg39 has also been reported as a nucleophagy receptor [113]. It was shown that Atg39 is a single-pass membrane protein and the *N*- and C-terminal regions are exposed to the cytoplasm and perinuclear space, respectively. Atg39 is anchored to the outer nuclear membrane (ONM) via a single-pass membrane domain, whereas it is associated with the inner nuclear membrane (INM) via the amphipathic helices of the C-terminal [122]. The Atg8 and Atg11 binding sequences are contained in the *N*-terminal region [113]. The formation of nucleus-derived double-membrane vesicles (NDVs) is the key step in nucleophagy. The outer and inner membranes of NDVs are derived from the ONM and INM [113]. When nucleophagy was induced, the amphipathic helices of Atg39 not only promoted the assembly of Atg39 at the NDV formation site, but were also involved in the deformation of the nuclear envelope and generated short protrusions, the tips of which provided a site for NDV formation [120]. Then, Atg39 interacted with Atg11 that served as a scaffold to recruit ATGs and Atg8 located at the expanding IM, to load NDVs to the IM [113,122].

### 6.3. Xenophagy

The process by which invading pathogens are encapsulated by double-membrane vesicles (DMVs) and transported to the lysosome for clearance is called xenophagy. This process increases the antigen presentation of cells and induces a further immune response. Therefore, xenophagy also provides a broad spectrum of defense mechanisms to capture bacterial, viral, and protozoan pathogens [123]. However, these pathogens have evolved different ways to escape from or paralyze xenophagy.

It was found that SopF, as a *Salmonella* T3SS effector, blocked bacterial autophagy [124]. In wild-type *Salmonella* cells, the bacterial-containing vacuolar membrane curvature changed, and the vacuolar membrane was damaged. V-ATPase, which is located in the membrane, recruited ATG16L1 to the vacuole to initiate LC3 lipidation. However, SopF targets ATP6V0C in the V-ATPase for ADP-ribosylation. Therefore, the V-ATPase could not bind to ATG16L1, and the lipidation of LC3 was restrained. Thus, xenophagy was blocked (Figure 4a).

In addition to blocking xenophagy, there is another mechanism by which xenophagy is escaped. If a pathogen cannot be identified, the cells cannot respond to xenophagy. It was found that TMEM41B is essential for DMV formation during β-coronavirus infection [125]. The β-coronavirus family includes SARS-CoV, SARS-CoV-2, MERS-CoV, and MERS-CoV. β-coronaviruses change membrane curvature and reshape host cell endomembranes to form DMVs for genome replication and transcription. After they enter host cells, the genomic RNA is translated into nonstructural proteins (nsp1-16). These polyproteins such as nsp3-nsp4 are cleaved into individual proteins by viral genome-encoded proteases. TMEM41B facilitates the rearrangement and binding of nsp3 and nsp4, leading them to the two sides of the ER. The interaction and separation of nsp3/4 induced the bending of the ER to form membrane curvature and further formed DMVs that escaped xenophagy [125,126] (Figure 4b). 

## 7. Discussion

The change of membrane curvature is one of the eternal themes of biofilm systems, which play a key role in many biological processes. It is a dynamic process with precise regulation. Therefore, how to ensure various reactions occur at the right time at a suitable speed in membrane extension merits discussion. Currently, there are many membrane sources reported for IM extension, including various organelles with membrane structure, such as the ER, Golgi, mitochondria, and nucleus. However, the membrane sources still remain to be fully explored. Exosomes are extracellular vesicles originating from endocytosis and are critical messengers involved in cell-to-cell communication [127]. They also alter the tumor microenvironment, growth, and progression by orchestrating various autocrine and paracrine functions [128]. Therefore, cell structures with membranes, such as exosomes may be potential membrane sources for the IM. In the process of autophagosome formation, the changes of membrane curvature accompany the entire event from autophagy initiation to vesicle closure. However, the fusion mechanism between the autophagosome and lysosome is rarely reported. Several SNARE proteins have been reported to be involved in the fusion, such as STX17 and SNAP29 [129], but little is known about the contribution of the lysosome contents or the real-time status and stimulations of the fusion process. The acidic pH of the lysosome is maintained by V-ATPase and is associated with cell death via lysosome membrane permeabilization [130]. Whether the acidic environment of the lysosome affects the membrane curvature during the fusion process should be verified. The scaffold proteins mediating autophagy membrane curvature usually superimpose on the membrane by electrostatic interactions, whereas the transmembrane proteins insert into the membrane by amphiphilic helices in membrane organelles. However, it was reported that Epr1 is a soluble protein and mediates the association between Atg8 on the IM and the integral membrane proteins VAPs on the ER to contribute to ER-phagy [131]. Therefore, it would also be of value to explore new proteins beyond membrane proteins and the mechanisms that mediate membrane curvature and autophagy activation in the future.

## Figures and Tables

**Figure 1 cells-12-01132-f001:**
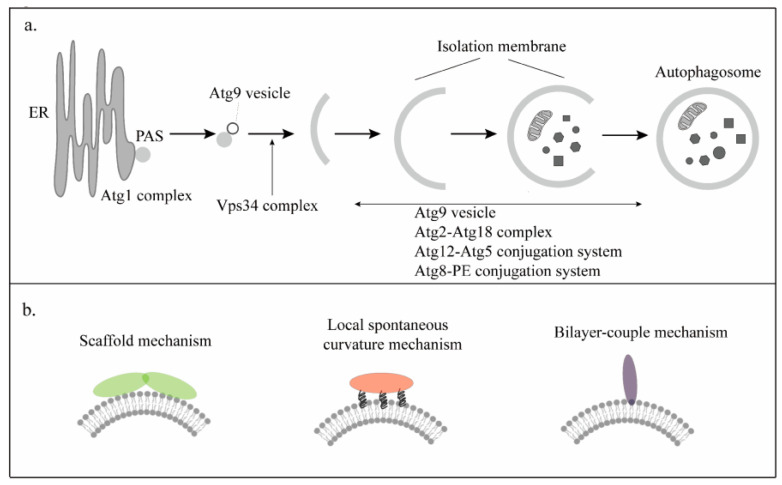
The complexes of autophagy, and the different mechanisms of membrane curvature. (**a**) There are six complexes in autophagy, namely the Atg1 complex, Vps34 complex, Atg2-Atg18 complex, Atg12-Atg5 conjugation system, Atg8-PE conjugation system, and Atg9 vesicle. (**b**) The mechanisms of membrane curvature. Left: The scaffold mechanism. The proteins function as scaffolds that present an intrinsic curvature. Middle: The local spontaneous curvature mechanism. The amphiphilic helices of the protein are embedded in the lipid matrix. Right: The bilayer-couple mechanism. The amphipathic protein domains penetrate only one lipid monolayer.

**Figure 2 cells-12-01132-f002:**
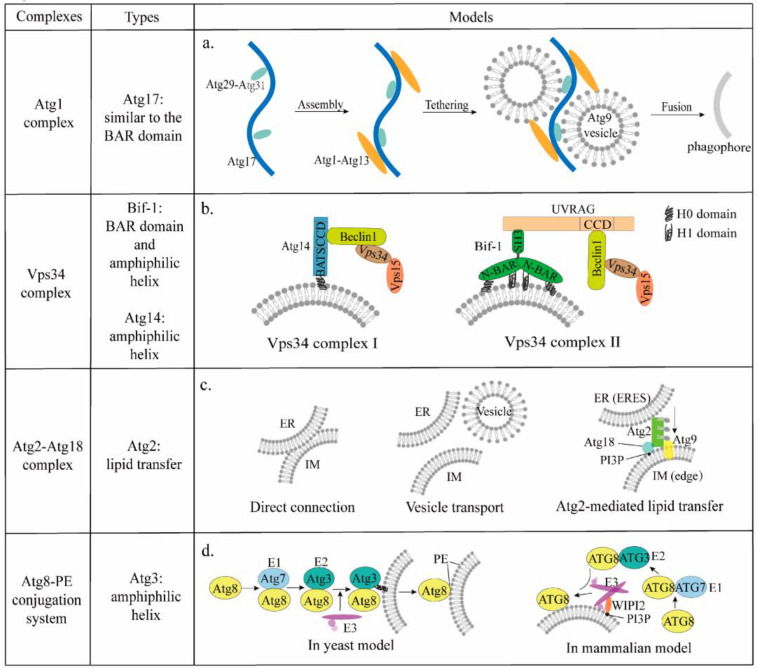
The different types and models to explain the mechanisms inducing membrane curvature. (**a**) Atg17 of the Atg1 complex senses membrane curvature and tethers Atg9 vesicles through its BAR-like domain. (**b**) Left: Atg14 recruits Beclin1 to produce PI3P to promote the autophagic process. Right: Bif-1 of the Vps34 complex affects autophagy by tethering Atg9 vesicles through the BAR domain and amphiphilic helices. (**c**) The Atg2-Atg8 complex builds a pathway for lipid transport from the ER to the IM. (**d**) In the yeast model, Atg3 targets the IM through its amphiphilic helix and then catalyzes Atg8 lipidation to complete the autophagic process. In the mammalian model, WIPI2 recruits ATG12-ATG5-ATG16 which covalently links ATG8 to phosphatidylethanolamine lipid headgroups in the phagophore membrane to the phagophore.

**Figure 3 cells-12-01132-f003:**
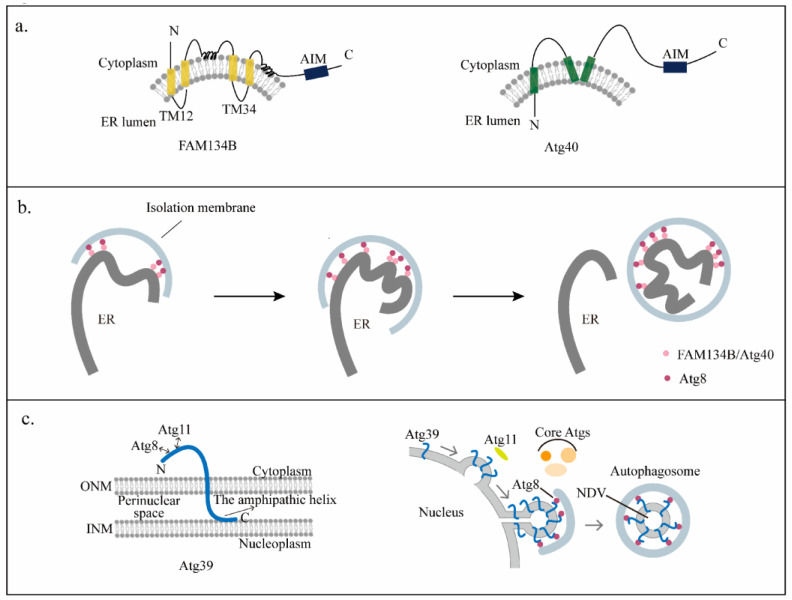
The models of ER-phagy and nucleophagy. (**a**) The structure of FAM134B and Atg40. Left: TM12 and TM34 anchor to the ER membrane, and two amphipathic helices position on the membrane on either side of TM34 to induce membrane curvature. Right: There are three transmembrane domains that like the reticulon domain induce the production of highly curved ER fragments. (**b**) A model for local ER remodeling by FAM134B/Atg40 in ER-phagy. When IM formation initiates in the vicinity of the ER, FAM134B/Atg40 induces membrane curvature, and then the AIM domains of FAM134B/Atg40 interact with Atg8 on the IM to wrap these fragments into the IM, leading to efficient ER packing of the IM. (**c**) Atg39 is anchored to the ONM via the single-pass membrane domain, and it is associated with the INM via the amphipathic helices of the C-terminal. When nucleophagy is induced, the amphipathic helices of Atg39 promote the assembly of Atg39 at the NDV formation site, and are also involved in the deformation of the nuclear envelope.

**Figure 4 cells-12-01132-f004:**
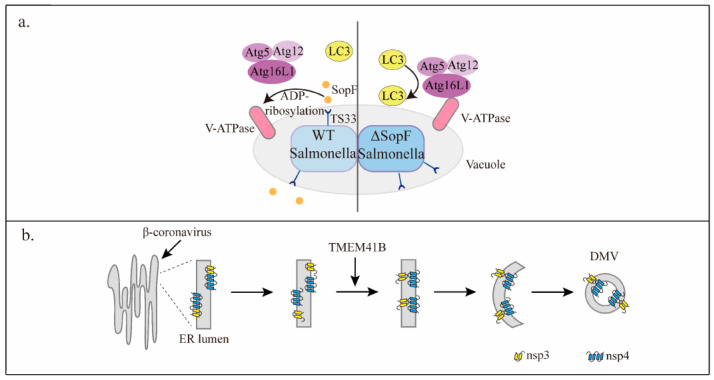
The different mechanisms of escape from xenophagy. (**a**) When cells are infected by *Salmonella*, the membrane curvature of the vacuole changes and the vacuolar membrane is damaged. SopF targets V-ATPase for ADP-ribosylation. The V-ATPase cannot recruit ATG16L1 to the vacuole to initiate LC3 lipidation to escape from xenophagy. (**b**) When cells are infected by β-coronavirus, the β-coronavirus is translated into nsp. The rearrangement and binding of nsp3 and nsp4 induce the bending of the ER lumen to form a membrane curvature and further form DMVs to escape xenophagy.

## Data Availability

Not applicable.

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
