# Peer review of "Membrane Curvature: The Inseparable Companion of Autophagy"

_cells, 2023, doi:10.3390/cells12081132_

Round 1

Reviewer 1 Report

The review by Liu and colleagues on the role of membrane curvature in autophagy pathway provided a detailed review about this feature in the budding yeast S.cerevisiae and mammals. However, there are some parts of the review that are not accurate with the published information and the citation of the literature is not always complete or balanced. I have a series of comments at helping to ameliorate and make the review more complete and precise:

Along the manuscript, some terminology is not properly used. For instance, the nucleation or elongation occurs in the phagophore or isolation membrane, but not in the autophagosome (the complete double-membrane vesicle). In S. cerevisiae, autophagy pathway starts at the phagophore assembly site or pre-autophagosomal structure (PAS), which leads to the formation of the phagophore or isolation membrane (IM).

There are some sentences mentioned along the manuscript without any supporting literature. It would be good that authors insert corresponding references after this information. Examples: Lines 53, 201, 250, 284.

Specific comments:

- Line 26. In the abstract, a BAR domain is mentioned, but not from which protein.

- Lines 29-33. Confusing sentences for the reader, not accurate or well explained.

- Lines 43-45. In this fragment of the manuscript, authors claim that each shape of the endoplasmic reticulum determines the role within the ER. They should consider to include bibliography that sustain this notion.

- Line 54. Authors introduce autophagy pathway, however they only focus in mammals. I would recommend that they introduce as well the differences with, at least, S. cerevisiae model, since latter it would be extensively explained.

- Line 78. In Figure 1, authors suggest that the Atg1 complex is initially located at the ER membrane, prior the presence of Atg9 vesicles. Until now, there is no data holding this statement. In addition to that, the PAS does not present a curvature structure upon its initiation (Fujioka et al, Nature, 2020).

- Line 126. In the section 3 (membrane curvature during initiation phase of autophagy), it would be pertinent to discuss about the membrane curvature of the Atg9 vesicles.

- Line 170. In the Figure 2, there are some confusing drawings. In Figure 2a, the Atg1 complex is not precisely described as shown in literature (Ragusa et al, Cell, 2012; Rao et al, Nat Comm, 2016). Additionally, as shown in the Figure 2b, it seems that PI3K complex I and II are localized in opposite places within the same membrane, however no data support this drawing. In Figure 2d, if talking about Atg8-family proteins, I would recommend to mention as Atg8 instead to LC3.

- Line 181. It’s recommended that authors explain how interaction between Atg17 and Atg29 happens, with corresponding citations.

- Line 188. This paragraph is only talking about regulation of yeast Atg13, and it should be reflected like this. It would be important as well to explain how human Atg13 is regulated as well.

- Lines 202-203. I would suggest to have more elaborate text: which are the other regulatory signals, and which are the other sources of autophagosomal membrane?

- Line 208. Mix of yeast and human ATG names, they should be correctly mentioned as Atg14/ATG14L and Vps38/UVRAG. In addition, some literature should be added: Matsunaga et al, Nat Cel Biol, 2009; Zhong et al, Nat Cel Biol, 2009.

- Line 262. Authors mentioned a hypothetical model about Bif-1 and Atg9 vesicles. They should elaborate better this part, based on the manuscript published afterwards by the same group (Takahashi et al, Autophagy, 2009).

- Line 283. This reference is missing: Matsunaga et al, J Cel Biol, 2010.

- Line 298. Missing information about Atg2: it is well known that this protein, together with Atg9 and Atg18, is located at the edges of the phagophore membrane (Graef et al, MBoC, 2013; Suzuki et al, J Cel Sci, 2013), which are highly-curved regions, but also preferentially associates with membranes carrying lipid-packing defects (Gómez-Sánchez et al, J Cel Biol, 2018), and contains an amphipathic helix (Kotani et al, PNAS, 2018). N-terminal domain of Atg2 is similar to the one of Vps13, but they’re not binding between each other (as it referred in the text), and there is no supporting data showing that this domain localizes at the ER exit sites. It has been shown that Atg2/ATG2 is required to establish membrane contact sites between the phagophore and the ER (Gómez-Sánchez et al, J Cel Biol, 2018; Kotani et al, PNAS, 2018; Valverde et al, J Cel Biol, 2019), and this is, at least, dependent on the interaction between Atg2/ATG2A and Atg9/ATG9A (Gómez-Sánchez et al, J Cel Biol, 2018; Tang et al, Cell Rep, 2019).

- Lines 305-306. What authors want to describe when talking about “separatist membrane”?

- Line 308. Recently, it has been shown that Atg9/ATG9A is a scramblase protein, allowing the asymmetry in the lipid gradient (Matoba et al, Nat Struc Mol Biol, 2020; Maeda et al, Nat Struc Mol Biol, 2020; Ghanbarpour et al, PNAS, 2021; Chumpen Ramirez et al, Autophagy, 2022).

- Line 310. Important recent advances about membrane curvature in ATG16L and WIPI2 have been discovered, and they should be included in the review: Jensen et al, Sci Adv, 2022.

- Line 323. Missing references: Kabeya et al, EMBO J, 2000; Kirisako et al, J Cel Biol, 2000; Ichimura et al, Nature, 2000.

- Line 324. Missing information about Atg3/ATG3: Hervás et al, Sci Rep, 2017; Want et al, Prot Sci, 2020; Ye et al, Nat Comm, 2021.

- Line 352. Missing information: Mochida et al, Nature, 2015; Cui et al, Science, 2019.

- Line 346. Besides ER-phagy and xenophagy, there are more selective types of autophagy driven by membrane curvature. It would be recommended that authors elaborate some text about other selective degradative events, such as nucleophagy.

- Line 410. Authors should rewrite the conclusion and bring the relevance of membrane curvature in autophagy pathway. As it is now, it is simply a resume of what has been described along the review.

Citations should be well placed, not mentioning the name of the first author, but rather “surname and colleagues”. Examples: Lines 239, 326…

Several typographical mistakes were throughout the manuscript. E.g. DFCP1 (line 269), Atg8 (line 324), helices…

Reviewer 2 Report

The paper by Liu et al. describes the process of membrane curvature during the different autophagy steps. This review is of interest; however I think it requires major modifications in terms of form and content prior its publication.

There are recent papers that need to be included and discussed: PMID 36516251, 36122245, 35332264, 33826365.

The conclusion section need to be re-written. Currently the conclusion section is manly a summary of the previously reported data, and a real discussion of the presented studies is lacking. I suggest concluding to state the relevance of this process and evaluate whether changes in membrane curvature or membrane curvature proteins are involved in any pathological situation. This would differentiate this paper from previous published revision papers that addressed the same subject. A “future perspectives” section containing the unanswered questions in the area should also be added.

In the references, please reduce the number of cited review and add the original papers.

The review also requires an improvement from a formal point of view.

Please add the complete official protein name the first time proteins are mentioned (ej line 142). Authors also write about “Complex II”, but they do not indicate what this refers to.

The quality of the figure is really low, please be sure that they are uploaded with a higher resolution.

Michal J, ref 28, should be referred to as Ragusa M. et al. as Michael is its first name, the same for reference 44, this continues throughout the whole text.

The text should be revised by an English native speaker, there are sentences that are unclear (ej: “ Starvation is responded to Complex  I at autophagosome”) and words that do not make sense (ej: Separatists membrane), I also suggest to eliminate colloquial sentences such as “and so on”, “in many different ways”.

Round 2

Reviewer 1 Report

Dear authors,

After your revision, I think you fulfilled all the doubtful/incomplete information along the manuscript

Author Response

Dear editors and reviewers,

Thank you so much for your reviewing! We deeply appreciate your recognition of our manuscript.

Best wishes,
From all authors

Reviewer 2 Report

Authors considered my previous comments; however the Text is still difficult to read and needs to be revised by a Native English Speaker. The reported information is valuable and interesting, but the form it is presented is still difficult to follow. I think the whole text needs to be edited to be suitable for publication.

There are corrections that I do not understand. For example, in line 65 it is now written that the phagophore is digested in lysosomes. Based on my knowledge this is not correct, the phagophore first has to growth into the autophagosome which then fuses to the lysosome for cargo digestion, at least in mammals. The same in line 77. Authors have mixed information from yeast and mammals and the result in some sections is confusing.  Please revise this throughout the whole text.
